# Validation of the Hungarian Version of the International Consultation on Incontinence Modular Questionnaire on Female Lower Urinary Tract Symptoms (ICIQ-FLUTS)

**DOI:** 10.3390/jcm13237389

**Published:** 2024-12-04

**Authors:** Wesam A. Debes, Munseef Sadaqa, Alexandra Makai, Olívia Dózsa-Juhász, Nikolett Tumpek, Judit Kocsis, Pongrác Ács, Réka Laura Szűcs, Zsanett Németh, Viktória Prémusz, Marta Hock

**Affiliations:** 1Doctoral School of Health Sciences, Faculty of Health Sciences, University of Pécs, 7624 Pécs, Hungary; munseef.sadaqa@etk.pte.hu (M.S.);; 2Institute of Physiotherapy and Sports Science, Faculty of Health Sciences, University of Pécs, 7624 Pécs, Hungary; 3Physical Activity Research Group, Szentágothai Research Centre, 7624 Pécs, Hungary; 4Dorottya Kanizsai County Hospital, 8800 Nagykanizsa, Hungary; 5Rátgéber Basketball Academy, 7624 Pécs, Hungary; 6National Laboratory on Human Reproduction, University of Pécs, 7624 Pécs, Hungary; 7MTA-PTE Human Reproduction Scientific Research Group, 7624 Pécs, Hungary

**Keywords:** ICIQ-FLUTS, lower urinary tract symptoms, questionnaire, validation study, women, incontinence

## Abstract

**Objectives:** Urinary incontinence (UI) is a prevalent condition that significantly impacts the quality of life. This study aimed to validate the Hungarian version of the International Consultation on Incontinence Questionnaire-Female Lower Urinary Tract Symptoms (ICIQ-FLUTS) and assess its psychometric properties in the context of the Hungarian population. **Study design:** A cross-sectional study involved 215 Hungarian-speaking women with a mean age of 67.6 ± 11.9 years. **Main outcome measure:** Participants were administered both the ICIQ-FLUTS and the International Consultation on Incontinence Questionnaire-Short Form (ICIQ-SF). The psychometric analysis included test–retest reliability, convergent validity, and internal consistency. **Results:** The Hungarian version of ICIQ-FLUTS demonstrated strong psychometric properties. The test–retest reliability analysis showed a high intraclass correlation coefficient (ICC = 0.921), indicating excellent agreement between measurements over a 14-day interval. Convergent validity was supported by a strong positive correlation between the total scores of ICIQ-FLUTS and ICIQ-SF (ρ = 0.686, *p* < 0.001), emphasizing shared underlying constructs. Furthermore, the ICIQ-FLUTS questionnaire exhibited good internal consistency, with a Cronbach’s α coefficient of 0.862. **Conclusions:** This study successfully validated the Hungarian version of the ICIQ-FLUTS questionnaire and demonstrated its robust psychometric properties. This tool will enable healthcare practitioners and researchers to effectively assess and address UI’s impact on their quality of life.

## 1. Introduction

Urinary incontinence (UI) is defined as “the complaint of any involuntary leakage of urine” [1]. UI might not be a life-threatening condition, but it has an undesirable influence on psychological and social life [2]. In addition, UI exerts a significant influence on one’s daily and social life, affecting aspects like employment, leisure travel, physical activity, and sexual well-being [3].

UI increases with age; however, across all age groups, the prevalence of UI is higher among females as compared to males [4,5]. It is worth mentioning that there is a huge variation in UI prevalence in different countries, ranging from 14% [6] in Mexico to 80% in Egypt [7]. In addition to age, UI is associated with multiple risk factors such as smoking, hypertension, diabetes, obesity, and number of deliveries [6,8].

In 2001, a Hungarian national survey was conducted to investigate urinary incontinence among women over the age of 18. Among 35,448 women who were asked about their related symptoms to urinary incontinence, 56% of participants had complaints of incontinence; also, 36% declared themselves to be incontinent [9]. In another survey conducted in 2011, 3506 women (33.9%) out of 10,403 reported that they had lost urine in their lifetime. In addition, urinary incontinence occurred rarely in 90.4%, 8.6% experienced UI several times a day, and constant symptoms of UI were reported by 1.0% of women. The symptoms usually increased with coughing/sneezing (23.7%) and with a strong urge to urinate (7.9%) among those who completed the questionnaire [10].

In a recent study that was conducted by Ambrus et al., the most prevalent type of UI was found to be stress incontinence with a prevalence of 64.9%, followed by mixed incontinence, urge incontinence, coital incontinence, and enuresis nocturna, at 21.1%, 10.5%, 7.2%, and 1.8%, respectively [11]. Furthermore, it is known that healthcare seeking for UI in Hungary is considered low, especially among women with milder symptoms and less knowledge about the condition [12].

The Third International Consultation on Incontinence recommended that randomized trials assessing treatments for incontinence should employ high-quality questionnaires, particularly the International Consultation on Incontinence Questionnaire (ICIQ), to evaluate their impact on patient outcomes and facilitate comparisons [13].

Discussing UI symptoms while taking clinical history could be challenging for clinicians due to society’s stigma [14]. In these cases, the self-completed questionnaire is considered a suitable tool to determine subjects’ perceptions about this condition and their therapeutic goals.

The International Consultation on Incontinence Questionnaire-Short Form (ICIQ-SF) has been developed and validated to assess urinary incontinence in the Hungarian language [15]. However, there is a need for a comprehensive and female-specific questionnaire that has been translated and validated for assessing overall LUTSs in Hungary.

The aim of this study is to validate and evaluate the psychometric properties of the Hungarian-language version of the International Consultation on Incontinence Questionnaire-Female Lower Urinary Tract Symptoms (ICIQ-FLUTS) for use in clinical practice and research.

## 2. Materials and Methods

At the start of the study, all participants were asked to fill out a questionnaire that was specifically created for this research. The questionnaire collected information about their background, including their age and education level, as well as details about the number of pregnancies, methods of deliveries, and other related factors.

### 2.1. Study Design

For this cross-sectional study, a convenience sample of 215 participants from both elderly homes and the community in the Southern Transdanubia Region, Hungary, was recruited from June 2021 to November 2022. Out of a total of 250 patients invited to participate in both paper-based and electronic questionnaire administration, 35 participants were excluded from the analysis due to rejection to participate or incomplete questionnaire responses. An adequate sample size was determined to be a ratio of 10 subjects to 1 variable. Therefore, a minimum of 120 women was required [16].

### 2.2. Data Collection

Participants completed the Hungarian version of the ICIQ-FLUTS, as well as a validated Hungarian version of the International Consultation on Incontinence Questionnaire-Short Form (ICIQ-SF). To fulfil the inclusion criteria, participants needed to be aged 50 years or above, regardless of whether they experienced LUTS, and all of them were Hungarian-speaking women. Per our exclusion criteria, the study did not include women experiencing mental disorders, neurological disorders, or severe visual impairments that hindered their ability to respond to questions.

### 2.3. Instruments

The ICIQ-SF comprises four inquiries designed to assess the frequency, severity, and impact of UI, as well as a series of eight self-diagnostic questions aimed at assessing the underlying causes or conditions leading to UI in patients. The questionnaire’s overall score ranges from zero to twenty-one, and it is categorized as follows: no impact (0 points), mild impact (1–3 points), moderate impact (4–6 points), severe impact (7–9 points), and very severe impact (10 or more points) [15,17].

ICIQ-FLUTS was established to be able to evaluate multiple characteristics of lower urinary tract symptoms (LUTSs) and their impact on quality of life; it was derived from the Bristol Female Lower Urinary Tract Symptoms (BFLUTS) questionnaire [18]. The ICIQ-FLUTS questionnaire is composed of 12 inquiries that evaluate lower urinary tract symptoms (LUTSs), and it is categorized into three domains: filling (four questions), voiding (three questions), and incontinence (five questions) [13]. Each item is scored from 0 to 4, and the total score varies between 0 and 48; higher scores are interpreted to represent the worst condition. In addition, ICIQ-FLUTS has a scale of bother from 0 to 10 following each item; this scale is not included to calculate the overall score, but it can be used as an indicator to measure the effect of symptoms on patients’ QoL.

Prior to commencing our research, we obtained permission from the ICIQ group. After obtaining their permission, we conducted the Hungarian version of ICIQ-FLUTS, which is accessible on the ICIQ group’s official website.

### 2.4. Translation

The ICIQ group confirmed that the translation process of the questionnaire followed the following stages: Initially, bilingual native speaker(s) of the Hungarian language performed the translation. Subsequently, the translated questionnaire was back-translated into English by bilingual native English speakers who were not involved in the initial translation phase. The back-translated version underwent a comprehensive review, including an assessment by the ICIQ group. Adjustments were made as necessary to ensure accuracy and consistency. To further validate the questionnaire, cognitive interviews were conducted with the target population by bilingual interviewers. Any unresolved conceptual equivalence issues identified during the interviews were carefully reviewed by the ICIQ group, and appropriate adjustments were made accordingly. These four steps align with the guidelines for the cross-cultural adaptation of self-reported measures, as documented in the existing literature [19].

### 2.5. Psychometric Analysis

To assess the stability of the questionnaire, a repeated-measures design (test re-test) was employed where a total of 34 participants completed the same questionnaire twice, with an interval of 14 days between the evaluations. The clinical condition of LUTSs was expected to remain stable during this period, thereby enabling the examination of the questionnaire’s consistency over time [16].

Internal consistency was tested to identify to which extent the various items within the questionnaire are interconnected or correlated with one another. Convergent validity was assessed by correlating the ICIQ-FLUTS scores with The International Consultation on Incontinence Questionnaire-Short Form (ICIQ-SF) score [15].

### 2.6. Statistical Analysis

Descriptive statistics were used to summarize the study population’s sociodemographic and clinical characteristics as mean ± SD and frequencies (%). We examined data normality using the Kolmogorov–Smirnov test (considering data as normally distributed when the *p*-value was > 0.05).

The assessment of floor and ceiling effects, which can affect the precision of data, involved calculating the proportion of patients who attained the maximum or minimum achievable scores. The presence of floor and ceiling effects was considered if over 15% of respondents reached either the highest or lowest possible scores [20].

The evaluation of internal consistency was executed through the utilization of Cronbach’s alpha coefficient statistical test [20,21]. Test re-test reliability was assessed by using intraclass correlation coefficients (ICCs) and Bland–Altman plots [20,22].

To evaluate the convergent validity, the Spearman Correlation Test was utilized due to the non-normal distribution of the data [23]. Data were entered into Microsoft Excel and analyzed by using SPSS version 23 (IBM Corp., Armonk, NY, USA). The analysis was carried out and reported according to the COSMIN reporting guidelines for studies on the measurement properties of patient-reported outcome measures [24].

## 3. Results

### 3.1. Study Participants

The study comprised a sample of 215 women with a mean age of 67.6 years ± 11.9. Ages ranged from 50 to 97 years. The mean body mass index (BMI) of the participants was 28.03 ± 12.09 kg/m^2^. The mean number of pregnancies reported by the participants was 1.94 ± 0.98. Furthermore, the participants had a mean of 1.90 ± 0.90 vaginal deliveries and a mean of 0.47 ± 0.79 cesarian deliveries. Table 1 shows additional characteristics of the volunteers in this study.

### 3.2. Internal Consistency

The internal consistency analysis demonstrated strong reliability for the entire questionnaire (α = 0.862), indicating consistent interrelatedness among the items. Cronbach’s alpha, if each item was deleted, consistently exceeded 0.80; detailed results can be found in Table 2.

### 3.3. Test Re-Test

The analysis of the test–retest reliability of the ICIQ-FLUTS questionnaire revealed a high level of stability for the total scores of urinary incontinence over the two-week period. The assessment of test–retest reliability involved the computation of the intraclass correlation coefficient of all questions (ICC = 0.921). As presented in Table 3, the test–retest values were examined for the complete questionnaire, as well as for each individual item and the subscales. Figure 1 illustrates the Bland–Altman plot for the test–retest reliability of the ICIQ-FLUTS.

### 3.4. Convergent Validity

The results indicated a strong positive correlation between the total scores of the ICIQ-FLUTS and ICIQ-SF (ρ = 0.686) (*p* < 0.001).

### 3.5. Floor and Ceiling Effect

We did not observe any floor and ceiling effect for the total score, as 11 participants (5.11%) reported the lowest possible score, and none reported the maximum score. Regarding the subdomains, two of the Subdomains reported floor effects with 111 participants (51.60%) and 61 participants (28.40%). No ceiling effects were noticed in any of the subdomains.

## 4. Discussion

The widely recognized ICIQ-FLUTS questionnaire offers a comprehensive instrument for women to report LUTSs and evaluate their impact on quality of life. This questionnaire’s global usage and validation in various languages underscore its significance in clinical practice and research [25,26].

The Hungarian version of ICIQ-FLUTS has proven its superior psychometric properties, such as adequate convergent validity. The good and positive correlation coefficients indicate that higher scores on the ICIQ-FLUTS questionnaire are associated with higher scores on the ICIQ-SF questionnaire, demonstrating a shared underlying construct that is evaluating UI; a relatively higher correlation was found in the Brazilian Portuguese validation of the ICIQ-FLUTS [27] with (r = 0.836) compared to (ρ = 0.68) in our study. Similarly, the Urdu version of the ICIQ-FLUTS scored an r = 0.820 correlation with ICIQ-SF [28].

Furthermore, internal consistency, determined by Cronbach’s α coefficient, was 0.86, exceeding the ICIQ group’s recommended minimum acceptable value of 0.70 [29]. The Polish validation of ICIQ-FLUTS yielded a value of 0.89 [25], which is similar to our findings. This underscores robust item correlation, confirming the instrument’s reliability and internal consistency.

The overall test–retest reliability scores of all questions exhibited strong stability (ICC = 0.92). These results indicate the tool’s suitability for the consistent and reliable evaluation of urinary incontinence symptoms over time. Our results were close to the previous findings’ validations in Brazilian-Portuguese and Urdu, with 0.901 and 0.970, respectively [27,28].

Regarding the specific subscales (filling, voiding, and incontinence), the filling subscale showed good stability (ICC = 0.85), the voiding subscale demonstrated moderate stability (ICC = 0.69), and the incontinence subscale exhibited strong stability (ICC = 0.90). Our results were slightly different compared to those of the validation of the ICIQ-FLUTS into the Urdu language, where the ICC score for incontinence subscales was close (ICC = 0.98); however, filling and voiding ICC scores were higher, with ICCs of 0.99 [28]. These higher scores might be explained by the fact that they examined test–retest reliability with a gap of 1 week between the two evaluations compared to 2 weeks in our study. The previous literature suggested that insufficient time intervals might allow study participants to memorize their first answers and recommended a 2 weeks interval as a generally appropriate period to minimize recall bias while maintaining the stability of the construct being measured [30,31,32]. Moreover, in the Brazilian-Portuguese validation study, intra- and inter-rater reliability were tested for each item separately in addition to the total score of the questionnaire; however, the researchers did not report test re-test scores for ICIQ-FLUTS subscales [27].

Nevertheless, a few items exhibited lower Intraclass Correlation Coefficients. This variance could be related to the intricate nature of symptom subtleties that each item attempts to capture within the context of lower urinary tract symptoms (LUTSs). It is essential to consider these results in a comprehensive manner. In addition, variations in the stability scores of the test–retest for different items could arise from a complex mix of factors impacting different aspects of LUTS. Considering the many facets of these symptoms, it is reasonable to think that certain items might show varying responses over time due to differences in personal, physical, and situational factors.

In our study, the total score did not exhibit floor or ceiling effects, but two of the three subdomains showed floor effects, affecting 51% and 28% of participants. This is contextualized within our study population properties, which include individuals with and without lower urinary tract symptoms (LUTSs). These floor effects appear clinically meaningful, indicating the instrument’s sensitivity to differences among respondents with varying LUTS-related health conditions. Additionally, the absence of a floor effect in the total score reaffirms the instrument’s effectiveness in assessing the overall status of lower urinary tract symptoms.

However, the Polish version of the ICIQ-FLUTS [25] observed no evidence of floor or ceiling effects within the group of individuals with lower urinary tract symptoms (LUTSs). Yet, when examining the control group, they identified a floor effect in both the total score and all subdomains. These differences in findings could be attributed to disparities in sample characteristics and variations in study designs, particularly the greater homogeneity that might present within each of their groups (LUTSs and control). In contrast, our study featured a more heterogeneous sample, as we did not segregate participants into distinct groups based on their LUTS status, which may account for these differences. We acknowledge, however, that to confirm these assumptions, future studies should compare these results with gold-standard measures, ensuring that participants who obtained a score of zero did not, in fact, experience any symptoms.

The instrument discussed in this article represents the first validation in the Hungarian language for a comprehensive assessment of lower urinary tract symptoms (LUTSs) in women. With the ICIQ-FLUTS, Hungarian-speaking women can now undergo a thorough evaluation of their LUTSs. Additionally, these tools allow for comparisons in research across various locations within Hungary, not to mention that early identification of symptomatic individuals through user-friendly screening tools is crucial for the timely treatment of female lower urinary tract symptoms (LUTSs), ultimately leading to improved health outcomes and quality of life.

However, it is essential to acknowledge certain limitations. The observation of a floor effect in specific questionnaire subdomains and variations in item-level correlations emphasizes the importance of conducting a focused examination using a comparative study design to evaluate both disease progression and response to treatment to ensure the generalizability of our findings.

## 5. Conclusions

To sum up, our study successfully validated the ICIQ-FLUTS for the Hungarian language. The questionnaires demonstrated strong convergent validity, internal consistency, and test–retest reliability, making it a convenient tool for diverse healthcare practitioners in clinical, primary healthcare, and epidemiological research to effectively address female lower urinary tract symptoms (LUTSs) in Hungary, which will eventually result in a better overall understanding of this disease and help with improving the clinical practice.

## Figures and Tables

**Figure 1 jcm-13-07389-f001:**
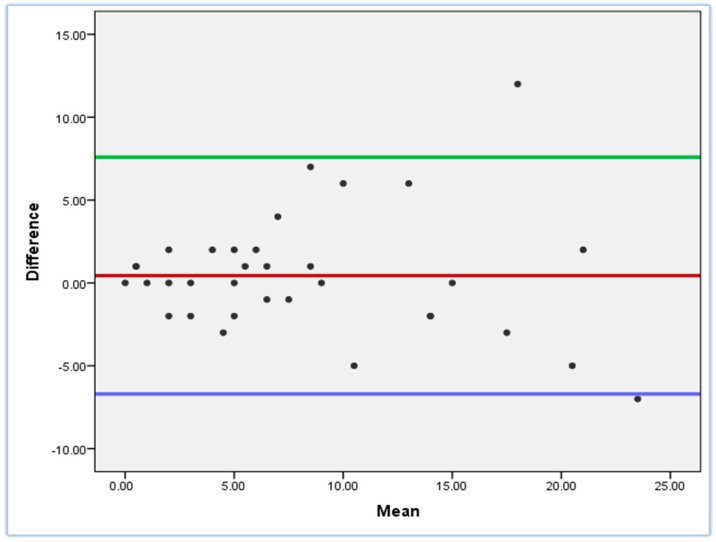
Bland-Altman plot.

**Table 1 jcm-13-07389-t001:** Demographic characteristics of participants (N = 215).

	n	Percentage
**Marital status**		
Married	91	42.50%
Not married	124	57.50%
**Education level**		
Primary school	77	35.80%
High school	67	31.20%
University degree	71	33.00%
**History of incontinence in the family**		
Yes	53	24.70%
No	162	75.30%
**Episiotomy**		
Yes	125	58.10%
No	90	41.90%
**Hysterectomy**		
Yes	41	19.10%
No	174	80.90%

**Table 2 jcm-13-07389-t002:** Item–total correlation and Cronbach’s alpha.

	Corrected Item–Total Correlation	Cronbach’s Alpha If Item Deleted
Fluts-1	0.448	0.858
Fluts-3	0.588	0.848
Fluts-5	0.397	0.860
Fluts-7	0.445	0.857
Fluts-9	0.357	0.864
Fluts-11	0.400	0.860
Fluts-13	0.484	0.855
Fluts-15	0.737	0.837
Fluts-17	0.694	0.840
Fluts-19	0.703	0.839
Fluts-21	0.699	0.841
Fluts-23	0.529	0.852

**Table 3 jcm-13-07389-t003:** Hungarian version of ICIQ-FLUTS reliability: Test–Retest.

Questions	Intraclass Correlation	95% Confidence Interval (CI) Lower Bound–Upper Bound	Cronbach’s Alpha Coefficient
Question 2a	0.722	0.44–0.86	0.717
Question 2b	0.763	0.51–0.88	0.784
Question 3a	0.738	0.47–0.86	0.735
Question 3b	0.689	0.37–0.84	0.683
Question 4a	0.839	0.67–0.91	0.838
Question 4b	0.383	−0.24–0.69	0.378
Question 5a	0.900	0.79–0.95	0.897
Question 5b	0.897	0.79–0.94	0.903
Question 6a	0.777	0.55–0.88	0.772
Question 6b	0.359	−0.30–0.68	0.352
Question 7a	0.523	0.03–0.76	0.515
Question 7b	0.875	0.75–0.93	0.872
Question 8a	0.467	−0.08–0.73	0.460
Question 8b	0.923	0.84–0.96	0.922
Question 9a	0.872	0.74–0.93	0.870
Question 9b	0.663	0.33–0.83	0.675
Question 10a	0.761	0.52–0.88	0.773
Question 10b	0.794	0.59–0.89	0.794
Question 11a	0.907	0.81–0.95	0.907
Question 11b	0.842	0.68–0.92	0.842
Question 12a	0.769	0.53–0.88	0.764
Question 12b	0.934	0.86–0.96	0.934
Question 13a	0.776	0.55–0.88	0.785
Question 13b	0.792	0.58–0.89	0.798
F score	0.852	0.70–0.92	0.849
V score	0.685	0.36–0.84	0.679
I score	0.903	0.80–0.95	0.906
Total score	0.921	0.84–0.96	0.920

## Data Availability

There are no linked research data sets for this paper. Data, a full version of the questionnaire, and scoring will be made available on request from the corresponding author.

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
