# Peer review of "Validation of the Hungarian Version of the International Consultation on Incontinence Modular Questionnaire on Female Lower Urinary Tract Symptoms (ICIQ-FLUTS)"

_jcm, 2024, doi:10.3390/jcm13237389_

Round 1
Reviewer 1 Report
Comments and Suggestions for Authors
Thank you for considering me as a reviewer for this publication. I have provided my comments as follows.
General Comments:
This article contributes to further progress in the evaluation of lower urinary tract symptoms, It is sometimes difficult to understand the problems associated with urination disorders, especially if the patient cannot explain the symptoms well enough. This is where the questionnaires come in handy.
Questionnaires are very useful in everyday medical practice and it is important that they are well designed but also approved by the appropriate international medical associations, such as ICS.
The authors explained in detail all the aspects associated with study design and acknowledged the limitations.
All the concepts of validation procedures are clearly defined.
It will be very useful for urologists, and other physicians that treat lower urinary tract symptoms.
Specific Comments:
Line 60-62: The sentence is a bit awkwardly written with a lot of parentheses. It seems to me that everything should be written together in one sentence with percentages in parentheses.
In a recent study that was conducted by Ambrus et al., the most prevalent type of UI 60 was Stress incontinence with [64.9%]. In addition, (Urge incontinence [10.5%]; mixed in-61 continence: [21.1%]; enuresis nocturna: [1.8%]; coital incontinence: [7.2%]) (11).
Author Response
Dear reviewer,
Please find attached the point-by-point responses to your comments.
Best regards

Reviewer 2 Report
Comments and Suggestions for Authors
This manuscript validates the Hungarian version of the ICIQ-FLUTS questionnaire, which assesses lower urinary tract symptoms (LUTS) among Hungarian-speaking women. The study is well-structured, employing an appropriate cross-sectional design with a robust sample size. The authors successfully demonstrate the questionnaire’s psychometric properties, including test-retest reliability, internal consistency, and convergent validity. However, the following comments must be addressed.
Lines 170 - 172: This paragraph appears to be some instructions from the manuscript template. It needs to be removed.
Line 202: Figure 1 is missing.
Line 237: How do the items that displayed lower Intraclass Correlation Coefficients compare with the other validated questionnaire versions?
Lines 314 and 315: Are there appendices to the manuscript that have not been provided? If there are none, the terms “Appendix A” and “Appendix B” must be removed.
Author Response
Dear Reviewer,
Please find attached the point-by-point responses to your comments.
Kind regards

Reviewer 3 Report
Comments and Suggestions for Authors
The manuscript submitted by Debes WA et al. is interesting and necessary for adequately evaluating Hungarian people with urinary incontinence. Urinary incontinence is a loss of bladder control that's commonly seen in older adults and women who have given birth or gone through menopause. The ICIQ-FLUTS is a questionnaire for evaluating the frequency, severity, and impact on quality of life (QoL) of urinary incontinence in men and women in research and clinical practice worldwide. This short and straightforward questionnaire is also helpful to general practitioners and clinicians in primary and secondary care institutions to screen for incontinence, obtain a comprehensive summary of the level, impact, and perceived cause of incontinence symptoms, and facilitate patient-clinician discussions. For a proper evaluation of the patient's symptoms, the doctors must provide a validated questionnaire in their language.
The manuscript is well organized - all the sections are competently presented. But there are some tipo mistakes:
l. 42, 43 the authors missed some points (.)
l. 46 Incidence of UI
l. 41, please delete what was conducted at the end of the sentence.
Subsection 2.6 needs more references
l. 170-172 should be deleted
I suggest the authors read the whole manuscript carefully again.
Comments on the Quality of English Language
The English needs minor revision.
l. 61-62 please reformulate, it is hard to understand
Author Response

(The authors gave the same response as above.)
